# Sick Leave Due to Ear Diagnoses, a Nationwide Representative Registry of Mexico

**DOI:** 10.3390/healthcare11081112

**Published:** 2023-04-13

**Authors:** Kathrine Jáuregui-Renaud, Ismael Velázquez-Ramírez, Jetzabel de Jetzenay Hernández-Tenorio, María del Carmen Solis-Cruz, Constanza Miriam Aguilar-Jiménez, Ofelia de Jesús Morales-Sánchez, Milliteotl Rincón-Rojas

**Affiliations:** 1Unidad de Investigación Médica en Otoneurología, Instituto Mexicano del Seguro Social, Ciudad de México C.P. 06720, Mexico; 2Coordinación de Salud en el Trabajo, Instituto Mexicano del Seguro Social, Ciudad de México C.P. 06720, Mexico; ismael.velazquez@imss.gob.mx (I.V.-R.); jetzabel.hernandez@imss.gob.mx (J.d.J.H.-T.); maria.solisc@imss.gob.mx (M.d.C.S.-C.); miriam.aguilar@imss.gob.mx (C.M.A.-J.); ofela.morales@imss.gob.mx (O.d.J.M.-S.); milliteotl.rincon@imss.gob.mx (M.R.-R.)

**Keywords:** vestibular, hearing loss, sick leave, work absenteeism

## Abstract

Sickness absence from work is a measure of both poor health and social functioning. In order to assess the frequency of sick leave due to ear-related diagnoses, we performed a retrospective analysis on the registry of paid sick leave certificates supplied by the main social security institution in Mexico during the years 2018 and 2019, just prior to the SARS-CoV-2 pandemic. We observed that, in the two years, 22,053 sick leave certificates due to ear-related diagnoses were provided to 18,033 workers. The most frequent ear-related diagnoses were those of vestibular disorders (94.64%); among them, the most common diagnosis was Benign Paroxysmal Positional Vertigo (75.16%), followed by Labrynthitis and Meniere’s disease (circa 8% each). A total of 4.63% of the diagnoses were related to external and middle ear disorders, and 0.71% were mainly related to hearing. Consistently, the highest cumulative days of sick leave required were given for the group of diagnoses related to vestibular disorders; although the less frequent diagnoses required the highest cumulative days per case (e.g., ototoxicity). During 2018 and 2019, the most frequent diagnoses of ear-related sick leave were due to vestibular diagnoses (particularly Benign Paroxysmal Positional Vertigo).

## 1. Introduction

Ear-related diagnoses are highly prevalent. According to the World Health Organization, nearly 20% of the global population lives with hearing loss [1]. Nonetheless, studies on the epidemiology of balance disorders in community-based adult populations have shown lifetime prevalence estimates for vertigo of 3% to 10% and for dizziness of 17% to 30% (for a review, see [2]). Data from a nationally representative sample of the United States of America from the Medical Expenditures Panel Survey 2007–2015 showed that among 221,273 patients over 18 years old, 5275 reported either vertigo or dizziness during that period, which were associated with high medical expenditures and utilization across various healthcare sectors [3].

Apart from its economic costs, sickness absence from work is a measure of both poor health and social functioning [4]. Although ear-related diagnoses can have consequences for work capacity, a systematic review of the literature showed few reports on the association between ear-related diagnoses and sick leave [5].

In México, in 530 consecutive patients referred to a specialized clinic (252 due to hearing loss and 278 due to vestibular disease), among those with paid employment, work absenteeism was reported by 5% of the patients with diagnoses related to hearing and 51% of the patients with vestibular disease, for a duration of 1 to 15 cumulative days and 1 to 365 cumulative days, respectively [6]. In England and Italy, in 400 consecutive patients with dizziness attending two neuro-otology clinics, 27% of the patients reported changing their jobs and 21% reported giving up work as a result of dizziness; the mean time of work absenteeism attributed to the dizziness in the previous 6 months was 7.15 days [7]. In the Netherlands, in 400 consecutive adult patients with paid employment attending a dizziness referral clinic, 244 patients (62%) reported that they had missed at least one part of a working day due to dizziness in the previous 4 weeks [8].

Contrariwise, long-term sickness absence due to dizziness/vertigo may have a low incidence. In a national follow-up study in Norway, the incidence of long-term sickness absence due to dizziness/vertigo was <1%, since one in four long-term sickness absentees with dizziness/vertigo obtained a disability pension [9]. In Sweden, a population-based prospective cohort study of 40,786 individuals who in 1985 were aged from 16 to 64 years and had a new sick absence of >7 days showed that the risk for future disability pension can be higher among patients initially on sickness absence due to oto-audiology diagnoses than among those due to non-oto-audiology causes [10].

The purpose of this study is to describe the frequency of sick leave certificates due to ear-related diagnoses that were provided by the main social security institution in Mexico (Instituto Mexicano del Seguro Social, IMSS) during the years 2018 and 2019, just prior to the SARS-CoV-2 pandemic.

## 2. Materials and Methods

A retrospective appraisal was performed on the nationally representative registry of paid sick leave of the main governmental organization that assists public health, pensions and social security in Mexico during the years 2018 and 2019. The registry includes all the certificates of sick leave that were previously validated by each of the decentralized administrations of health care and highly specialized hospitals of the institution, all over the country. During the years 2018 and 2019, the number of workers covered by work insurance was 19,623,022 in 2018 and 19,974,508 in 2019; each of these years, the institution certified circa seven million sick leave permissions related to general disease, covering circa 43 million days of sick leave related to general disease per year [11].

To determine the frequency of sick leave certificates due to ear-related diagnoses registered during the years 2018 and 2019, we identified those not related to work injuries, also removing duplicates and all the certificates related to maternity leave. Then, we quantified the ear-related certificates according to Chapter VIII of the International Classification of Diseases, 10th Revision (ICD-10) [12] by sex and group of age, as they were already classified in the registry.

According to diagnosis, the certificates and cumulative days of sick leave were classified in four groups: those related to the external and middle ear, those related mainly to hearing, those related to vestibular disorders and those less specific (Others).

The data are described according to their distribution by absolute numbers, percentages, and median and quartiles 1 and 3 (Q1–Q3), after the Kolmogorov–Smirnov test. Comparisons by sex were performed using *t*-test for proportions, with the significance level set at 0.05.

## 3. Results

Table 1 shows the frequency of sick leave certificates related to each of the ear-related diagnoses identified in the national registry of the IMSS, during the years 2018 and 2019. In the two years, 22,053 sick leave certificates due to ear-related diagnoses were given to 18,033 workers (10,447 women and 7586 men), 8563 workers during the year 2018 and 9470 workers during the year 2019. The frequency of certificates given by age was: 17.42% for workers between 18 and 30 years old, 25.93% for workers between 31 and 40 years old, 29.56% for workers between 41 and 50 years old, 22.48% for workers between 51 and 60 years old, and 4.58% for those older than 60 years.

In the two years, more certificates were given to women compared to men (Figure 1). However, the proportional number of women and the proportional number of men who required sick leave certificates were similar at almost any age, except for the fifth decade of life (from 41 to 50 years old), in which a larger proportion of women required sick leave certificates compared to men (32.5% of the women versus 25.6% of the men; *p* = 0.02).

During the two years, the most frequent diagnoses were those related to vestibular disorders (20,854 certificates; 94.64%), mostly Benign Paroxysmal Positional Vertigo (75.16% of all), Labrynthitis (8.50%) and Meniere’s disease (8.12%) (Table 1).

In the two years, 13,936 workers required sick leave due to Benign Paroxysmal Positional Vertigo (98,966 cumulative days; median 2, Q1–Q3 = 1–3); 1515 workers required sick leave due to Labrynthitis (18,851 cumulative days; median 3, Q1–Q3 = 1–5); and 1116 workers required it due to Meniere’s disease (40,873 cumulative days; median 3, Q1–Q3 = 2–15). Of note, workers with vestibular neuritis were fewer (n = 409) but required 16,632 cumulative days of sick leave (median 15, Q1–Q3 = 7–29).

On the other hand, only 4.63% of the diagnoses were related to external and middle ear disorders; while 0.70% were related mainly to hearing loss, including conditions such as tinnitus (0.34%). The less frequent diagnoses were mastoiditis and related conditions (0.01%), degenerative and vascular disorders of the ear (0.02%) and ototoxic hearing loss (0.05%) (Table 1).

Table 2 shows the cumulative days of sick leave registered during the two years, by diagnosis and by year. Consistently with the frequency of the diagnoses, the cumulative days of sick leave required were the highest for diagnoses related to vestibular disorders (176,200 days). The average cumulative days per case by diagnosis are shown in Figure 2. The highest cumulative days per case were required by the less frequent diagnoses: ototoxic hearing loss (137 days), other disorders of the vestibular function (52 days) and degenerative and vascular disorders of the ear (47 days), while fewer cumulative days were required by the most frequent diagnoses: Benign Paroxysmal Positional Vertigo (6 days), Labrynthitis (10 days) and Meniere’s disease (22 days).

## 4. Discussion

The results show compelling evidence that vestibular disorders are the most frequent cause of paid sick leave due to ear-related diagnoses in a nationally representative registry in Mexico. The high frequency of sick leave related to vestibular diagnoses was mainly related to the high frequency of Benign Paroxysmal Positional Vertigo, Labrynthitis and Meniere’s disease, while the less frequent diagnoses were related to the greatest cumulative days of sick leave.

In a previous study at the same institution, both self-reported limitations to perform daily life activities and sick leave were much higher in patients with vestibular disease than in patients with hearing loss (169/278 patients and 71/139 workers with vestibular disease, *versus* 41/252 patients and 8/136 workers with hearing loss, respectively) [6]. However, in that study, the frequency of the specific diagnoses differed from those observed in this study. This difference was probably due to the data source, since the previous study at the IMSS was performed at a specialized outpatient clinic, while this report includes records from all levels of health care, from emergency services to specialized outpatient clinics.

Additionally, in the previous study at the IMSS, patients with hearing loss reported from 1 to 6 visits to the clinic per year, and patients with vestibular disease reported from 1 to 8 visits to the clinic per year [6], which is consistent with a recent review showing that patients with dizziness/vertigo may report up to 9.6 visits per year at the primary care provider and 7.2 visits at the specialist [13].

The high frequency of Benign Paroxysmal Positional Vertigo as the main cause of paid sick leave is in agreement with the high incidence of the disease, provoking episodic vertigo [14]. A systematic review showed that, in community-based adult populations, the reported one-year incidence estimates of Benign Paroxysmal Positional Vertigo were from 0.06% to 0.6% [2]. In Germany, a cross-sectional, nationally representative neuro-otology survey of the general adult population showed a lifetime prevalence of Benign Paroxysmal Positional Vertigo of 2.4%, a one-year prevalence of 1.6% and a one-year incidence of 0.6%, with higher incidence rates in women than in men [15].

Although Meniere’s disease was considerably less frequent than Benign Paroxysmal Positional Vertigo, the cumulative days of sick leave related to this diagnosis were the second highest among all ear diagnoses. A clinical follow-up study has shown that, due to the paroxysmal nature of the disease and its related disorders, patients with Meniere’s disease may require sick leave repeatedly, but the cumulative days of sick leave may decrease after comprehensive treatment [16].

The greatest cumulative days of sick leave per case were observed among the diagnoses of ototoxic hearing loss, other disorders of the vestibular function and degenerative and vascular disorders. This finding is consistent with the severity/progressive evolution of these diagnoses, and permanent damage related to ototoxicity [17]. Noteworthy, previous studies support the fact that sick leave due to hearing or tinnitus diagnoses is not frequent but tends to be long [4]. Additionally, a systematic review of the association between hearing loss and employment among adults indicates that adult-onset hearing loss is associated with unemployment [18]. Even more, in Norway, a population-based cohort study indicated that hearing loss is weakly associated with sick leave but with an increased risk of receiving a disability pension [19]. These evidences are consistent with a lower self-reported absolute and differential productivity in cases of reduced hearing ability in noise [20], and the notion that the risk for future disability pension is higher among patients initially on sick leave due to oto-audiological diagnoses than among those due to non-oto-audiological diagnoses [10].

## 5. Conclusions

In a nationally representative registry in Mexico, >20,000 new paid sick leave certificates due to ear-related diagnoses were provided during years 2018 and 2019, with vestibular diagnoses (particularly Benign Paroxysmal Positional Vertigo) being the most frequent cause of ear-related sick leave. However, the highest cumulative days of sick leave per case were related to ototoxic hearing loss, other disorders of the vestibular function and degenerative or vascular disorders of the ear.

## Figures and Tables

**Figure 1 healthcare-11-01112-f001:**
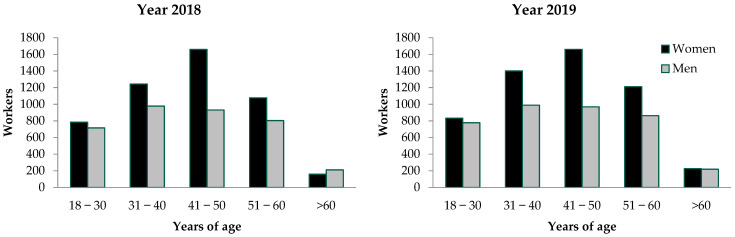
Absolute number of workers who required sick leave due to ear diagnoses during years 2018 and 2019, by age group, sex and year (either 2018 or 2019).

**Figure 2 healthcare-11-01112-f002:**
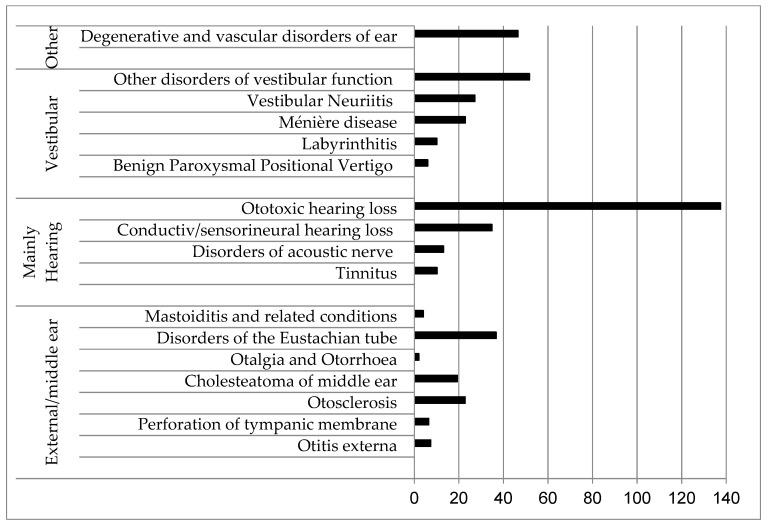
Average cumulative days of sick leave related to ear diagnosis during years 2018 and 2019; according to Chapter VIII of the International Classification of Diseases, 10th Revision [12].

**Table 1 healthcare-11-01112-t001:** Absolute number and frequency of new sick leave certificates related to ear diagnoses, according to Chapter VIII of the International Classification of Diseases, 10th Revision [12].

	ICD-10 Diagnosis	N	%
**External/Middle Ear**		**1022**	**4.63**
	Otitis externa	577	2.62
	Perforation of tympanic membrane	218	0.99
	Otosclerosis	80	0.36
	Cholesteatoma of middle ear	56	0.25
	Otalgia and Otorrhoea	55	0.25
	Disorders of the Eustachian tube	34	0.15
	Mastoiditis and related conditions	2	0.01
**Mainly Hearing**		**156**	**0.71**
	Tinnitus	75	0.34
	Disorders of acoustic nerve	38	0.17
	Conductive/sensorineural hearing loss	33	0.15
	Ototoxic hearing loss	10	0.05
**Vestibular**		**20,871**	**94.64**
	Benign Paroxysmal Positional Vertigo	16,575	75.16
	Labrynthitis	1875	8.50
	Meniere’s disease	1790	8.12
	Vestibular Neuritis	614	2.78
	Other disorders of vestibular function	17	0.08
**Other**		**4**	**0.02**
	Degenerative and vascular disorders of ear	4	0.02

Totals are highlighted in bold.

**Table 2 healthcare-11-01112-t002:** Cumulative days of sick leave related to ear diagnoses during years 2018 and 2019, according to Chapter VIII of the International Classification of Diseases, 10th Revision [12].

	ICD-10 Diagnosis	Cumulative Days 2018	Cumulative Days 2019	Total
**External/Middle Ear**				**9886**
	Otitis externa	3479	746	4225
	Perforation of tympanic membrane	732	665	1397
	Otosclerosis	1211	609	1820
	Cholesteatoma of middle ear	705	371	1076
	Otalgia and Otorrhoea	48	64	112
	Disorders of the Eustachian tube	157	1091	1248
	Mastoiditis and related conditions	4	4	8
**Mainly Hearing**				**3785**
	Tinnitus	164	603	767
	Disorders of acoustic nerve	246	250	496
	Conductive/sensorineural hearing loss	603	545	1148
	Ototoxic hearing loss	1371	3	1374
**Vestibular**				**176,200**
	Benign Paroxysmal Positional Vertigo	42,881	56,085	98,966
	Labrynthitis	9901	8950	18,851
	Meniere’s disease	19,168	21,705	40,873
	Vestibular Neuritis	8, 574	8, 058	16,632
	Other disorders of vestibular function	315	563	878
**Other**				**186**
	Degenerative and vascular disorders of ear	8	178	186

Totals are highlighted in bold.

## Data Availability

The data are contained within the article. The datasets are available from the corresponding author upon reasonable request.

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
