# Peer review of "Sick Leave Due to Ear Diagnoses, a Nationwide Representative Registry of Mexico"

_healthcare, 2023, doi:10.3390/healthcare11081112_

Round 1

Reviewer 1 Report

Thank you for the chance of reviewing your paper. It is interesting. I would suggest some improvments:
The diagram on page 108 shows the numver of men and women in two years. At least I can´t distinguish and interprete this diagram clearly. 
For me the incicidence of Menière´s disease seems to be rather high as well as the number of otoxic side effects. I would recommend to esplain these phenomenons in the discussion.

Author Response

Thank you for the chance of reviewing your paper. It is interesting. I would suggest some improvements:

  • We thank the reviewer for the useful recommendations.

The diagram on page 108 shows the number of men and women in two years. At least I can´t distinguish and interpret this diagram clearly.

  • In order to improve readability, the figure 1 has been redesigned

For me the incidence of Menière´s disease seems to be rather high as well as the number of otoxic side effects. I would recommend to explain these phenomenons in the discussion.

  • We thank you for the observation. The discussion now includes comments on the frequency of sick leave certificates that were provided due to these diagnoses. Since, independently from the incidence of the disease, they may imply limitations that provoke absenteeism to work.

Reviewer 2 Report

This descriptive epidemiological study provides insight into the impact of ear-related diagnoses on absenteeism from work. Overall, the manuscript is well written and needs only minor revisions. The following items should be addressed:

Introduction: page 2 line 56 - <1 what?

Results: page 2 line 87 - please check the totals. When I compare the total number in the text (22,053) to the table there is a discrepency. 

Discussion: page 5 lines 140-141 - means would be more useful (rather than the ranges that are reported)

Author Response

This descriptive epidemiological study provides insight into the impact of ear-related diagnoses on absenteeism from work. Overall, the manuscript is well written and needs only minor revisions. The following items should be addressed:

  • We thank the reviewer for all the valuable observations.

Introduction: page 2 line 56 - <1 what?

  • We apologize for the mistake. The symbol has been added.

Results: page 2 line 87 - please check the totals. When I compare the total number in the text (22,053) to the table there is a discrepancy.

-We thank the reviewer and we apologize for the errors. We have double checked the totals , and highlighted the corrections.

Discussion: page 5 lines 140-141 - means would be more useful (rather than the ranges that are reported)

  • We appreciate the comment. The new version clarifies the reason to use median and quartiles to describe variable data and absolute numbers to describe totals. 

Reviewer 3 Report

The abstract is disorganized and not concise. It does not allow to know the main objective of the study, its methodology, as well as the main findings obtained.
The introduction is well structured although most of the bibliographic citations are more than 5 years old.
The methodology is confusing and too brief, which makes it difficult to know how the study was proposed and how it was carried out, as well as the variables used.
The results are very poor, the authors limit themselves to a descriptive analysis. These data could provide more relevant information if statistical inference were made, comparing the main types of pathologies, for example, by sex and age.
In the discussion, only 4 new bibliographic citations are included in relation to the references included in the introduction, which is a great weakness.

The authors should include more bibliographic citations that allow discussion of the results obtained with those of other authors, in order to compare the data with those of other previous studies and improve the quality of the manuscript

Author Response

Reviewer 3

The abstract is disorganized and not concise. It does not allow to know the main objective of the study, its methodology, as well as the main findings obtained.

  • We apologize for the inconvenience. The abstract has been edited for clarity within the limits of the journal requirements.

The introduction is well structured although most of the bibliographic citations are more than 5 years old.

  • The references included were those available about sick leave or work absenteeism due to ear related diagnoses .

The methodology is confusing and too brief, which makes it difficult to know how the study was proposed and how it was carried out, as well as the variables used.

  • We thank the reviewer for the observation. The methodology section was expanded, according to the descriptive nature of the report.

The results are very poor, the authors limit themselves to a descriptive analysis. These data could provide more relevant information if statistical inference were made, comparing the main types of pathologies, for example, by sex and age.

  • The report has been classified as a Communication, in order to describe facts that were already recorded for different purposes. No hypothesis testing was intended, just communication of facts related to sick leave according to ear diagnoses, which may support further studies designed to test specific hypothesis.

In the discussion, only 4 new bibliographic citations are included in relation to the references included in the introduction, which is a great weakness. The authors should include more bibliographic citations that allow discussion of the results obtained with those of other authors, in order to compare the data with those of other previous studies and improve the quality of the manuscript.

-              We thank the reviewer for the observation. We were motivated to write this communication due to the scarcity of reports on work absenteeism related to ear-diagnoses. A new search of the literature provided a few more references related to the topic (though not specific) that are now included in the new version of the manuscript.

Reviewer 4 Report

Title: Sick Leave Due to Ear Diagnoses, A Nationwide Representative Registry of Mexico

In my opinion, this article needs to be improved and you should have done more statistical work.

Still I have important repairs to make:

1) The reference to the classification used in Table 1 "ICD-10 diagnosis" (page 2) is missing. 

2) The classification between "Otology" and "Audiology" is not congruent, because the pathologies that appear in "Otology" may also be included in "Audiology" as, for example, in "Condutiv/sensorineural hearing loss" which would be repeated. This classification is not understood, separating "Otology" from "Audiology". 

3) The numerical part of the Table is also not in agreement. Review the numbers of the "Vestibular" part because the sum of the components gives 20,871 and not 20,854.

4) Table 2:

The column of the total does not match. 

The 2nd row of the Table is swapped with the 3rd (in Total).

In the 7th row of the Table: 8 with 8 =16. Which causes the total not to match the sum. The "Tinnitus" line doesn't have the right numbers either.

In "Vestibular", also the 2nd and 3rd lines are wrong.

Author Response

Reviewer 4

In my opinion, this article needs to be improved and you should have done more statistical work.

  • We thank the reviewer for the comment. The manuscript was classified as a Communication, in order to describe facts that were already recorded for different purposes. We hope to support/orient the design of specific studies to test relevant hypothesis.

Still I have important repairs to make:

  • The reference to the classification used in Table 1 "ICD-10 diagnosis" (page 2) is missing.

- We thank the reviewer for this important observation. We have amended the two tables and Figure 2 with reference to ICD-19

2) The classification between "Otology" and "Audiology" is not congruent, because the pathologies that appear in "Otology" may also be included in "Audiology" as, for example, in "Condutive/sensorineural hearing loss" which would be repeated. This classification is not understood, separating "Otology" from "Audiology".

- We thank the reviewer for the valuable comment. Since we used the diagnoses described in the ICD-10 to identify the cause of sick-leave, to avoid confusion, we have edited the group labels used to present the data, all through the manuscript. 

3) The numerical part of the Table is also not in agreement. Review the numbers of the "Vestibular" part because the sum of the components gives 20,871 and not 20,854.

4) Table 2: The column of the total does not match.

- We thank the reviewer for the relevant comment, the Tables have been double checked in order to ascertain the totals.

The 2nd row of the Table is swapped with the 3rd (in Total).In the 7th row of the Table: 8 with 8 =16. Which causes the total not to match the sum. The "Tinnitus" line doesn't have the right numbers either. In "Vestibular", also the 2nd and 3rd lines are wrong.

  • We greatly appreciate these observations. The table has been completely revised including both numbers and format.